# Consideration of the cloud motion for aircraft-based stereographically derived cloud geometry and cloud top heights

Lea Volkmer[1], Tobias Kölling[1,*], Tobias Zinner[1], and Bernhard Mayer[1]

[1]Meteorologisches Institut, Ludwig-Maximilians-Universität München, Munich, Germany
[*]now at: Max Planck Institute for Meteorology, Hamburg, Germany

**Correspondence:** Lea Volkmer (L.Volkmer@physik.uni-muenchen.de)

**Abstract.** Cloud geometry and in particular cloud top heights can be derived from 2-D camera measurements by applying a stereographic method to data from an overflight over a scene of clouds (see e.g. Kölling et al., 2019). Although airplane overpasses are relatively fast, cloud motion with the wind is important and can result in errors in the cloud localization. Here, the impact of the wind is investigated using the method from Kölling et al. (2019) for measurements of the airborne hyperspectral imaging system specMACS (spectrometer of the Munich Aerosol Cloud Scanner). Further, a method for the cloud motion correction using model winds from the European Centre for Medium-Range Weather Forecasts (ECMWF) is presented. It is shown that the update is important as the original algorithm without the cloud motion correction can over- or underestimate the cloud top heights by several hundred meters dependent on the wind speed and the relative wind direction. This is validated using data from the EUREC$^4$A campaign as well as realistic 3-D radiative transfer simulations. From the comparison of the derived cloud top heights to the expected ones from the model input an average accuracy of the cloud top heights of less than $(20 \pm 140)\,\mathrm{m}$ (mean deviation and one standard deviation) is estimated for the updated method.

## 1 Introduction

The macro- and microphysical properties of clouds have a large impact on the Earth's radiation and energy budgets as they determine the cloud's interaction with solar and terrestrial radiation. Since clouds have a high spatial and temporal variability representing them accurately in both numerical weather prediction as well as global climate models is difficult, such that parametrizations of the cloud properties are important. Those often rely on observational studies which have been widely extended in recent decades. In particular, one aim has been to obtain a better understanding of the role of clouds in climate change. An important contribution to the characterization of cloud properties is provided by active and passive remote sensing instruments which can be ground-, aircraft- or satellite-based and sensitive to different wavelengths both in the solar and terrestrial spectral range. One of them is the airborne spectrometer of the Munich Aerosol Cloud Scanner (specMACS, Ewald et al. (2016); Pörtge et al. (2023); Weber et al. (2023)) operated onboard of the German High Altitude and LOng range research aircraft HALO (Krautstrunk and Giez, 2012). specMACS consists of two hyperspectral line cameras measuring radiances in the wavelength range between $400\,\mathrm{nm}$ and $2500\,\mathrm{nm}$ (Ewald et al., 2016) as well as two polarization resolving RGB cameras (Phoenix 5.0 MP Polarization Model) (Pörtge et al., 2023; Weber et al., 2023). That way, the solar radiation reflected from

clouds is measured with a high spectral and spatial resolution and a wide field of view ($32.7°$ and $35.5°$ for the line cameras (Ewald et al., 2016) and $91° \times 117°$ for the two polarization cameras combined (Weber et al., 2023)). The specMACS measurements are used for microphysical retrievals of cloud properties, both using a bispectral approach as described by Nakajima and King (1990) as well as from a polarimetric retrieval described in Pörtge et al. (2023), which uses multi-angle polarized radiance observations of the cloudbow to derived cloud droplet size distributions (CDSD). In general, passive observations usually do not provide cloud top height or structure with height resolution. However, for the interpretation and geolocalization of the microphysical retrievals, the exact location of the observed clouds in 3-D space is important. For that, the stereographic derived cloud geometry is used.

One approach for the localization of clouds in 3-D space is the stereographic reconstruction from multi-angle intensity measurements while flying over a scene of clouds. For example, this has been used for spaceborne instruments such as the Multi-angle Imaging SpectroRadiometer (MISR) (Moroney et al., 2002; Seiz and Davies, 2006) and the Advanced Spaceborne Thermal Emission and Reflection Radiometer (ASTER) (Seiz et al., 2006). For specMACS, the clouds are located in 3-D space using the 2-D intensity measurements of the polarization cameras. The algorithm is described in detail by Kölling et al. (2019) and is based on the identification of points on the cloud surface using contrast gradients. Afterwards, the points are re-identified in subsequent images and located in 3-D space using the multi-angle observation. Then, the derived cloud top heights are for example used for the geolocalization of the cloud targets observed by the polarimetric retrieval for the determination of the CDSD. The retrieval is based on the aggregation of the polarized radiance measured for a cloud target in subsequent images to obtain the full signal of the cloudbow, which is sensitive to both, the effective radius and variance of the CDSD. Thereby, accurate cloud top heights are important as small errors in the cloud top height assigned to cloud targets will lead to wrong localizations of the targets in subsequent images. Consequently, this leads to a wrong aggregation of the cloudbow signal and thus, errors in the derived CDSD (Pörtge et al., 2023).

So far, the stereographic retrieved cloud top heights were compared by Kölling et al. (2019) to the cloud top heights derived from the WALES lidar (WAter vapor Lidar Experiment in Space, Wirth et al., 2009) for simultaneous measurements during the NAWDEX (North Atlantic Waveguide and Downstream Impact Experiment) campaign in October 2016. They found a median bias of $126\,\mathrm{m}$ with the stereo heights found to be lower. It was indicated that the most prominent outliers in regions of high lidar cloud top height and low stereo height were observed for thin cirrus layers above cumulus clouds. In those scenes, the lidar is sensitive to the upper ice cloud layer while the stereo algorithm detects image areas with high contrasts which are preferably observed for lower cloud layers. Volkmer et al. (2023b) applied the stereographic reconstruction algorithm to synthetic measurements obtained from realistic 3-D radiative transfer simulations and found an average underestimation of the stereographic cloud top heights by $(70 \pm 130)\,\mathrm{m}$ without correcting for any cloud motion. Kölling et al. (2019) argued that in contrast to spaceborne methods employed e.g. to MISR (Moroney et al., 2002; Seiz and Davies, 2006) and ASTER (Seiz et al., 2006), no correction of the clouds wind movement needs to be applied. This is due to the much smaller time difference between two successive images (framerate of $1\,\mathrm{Hz}$ used for the stereo algorithm) and the much lower operating altitude leading to a much more rapid change in the observation angle compared to the satellite observations. However, the observation of the clouds from multiple viewing perspectives has been exploited to derive an estimate of the underlying 3-D wind field.

60 In subsequent evaluations of measurement data, the impact of the cloud motion on the stereographic derived cloud top heights has been further investigated by comparing the derived cloud top heights for flight legs which were flown forward and backward within a time difference of about 70 minutes. Deviations of several hundred meters despite the temporally highly resolved observation of the clouds from multiple viewing angles have been observed. This is due to the faster movement of the clouds through the field of view of the instrument when the aircraft is flying against the wind and leads to an overestimation

65 of the cloud top heights. Correspondingly, flying with the wind direction leads to an underestimation of the cloud top heights. Hence, an approach for the cloud motion correction using the 3-D wind field of the fifth generation European Centre for Medium-Range Weather Forecasts (ECMWF) atmospheric reanalysis (ERA5, Hersbach et al., 2020) has been developed and will be presented in Sect. 2 in this study. In Sect. 3, the cloud motion correction will be validated using measurement data. Finally, the approach will be validated using realistic 3-D radiative transfer simulations of the measurements performed with

70 MYSTIC (Monte Carlo code for the physically correct tracing of photons in cloudy atmospheres, Mayer, 2009; Volkmer et al., 2023b) in Sect. 4.

## 2 Wind correction based on ERA5 reanalysis data

To correct for the movement of clouds within the observation time, the ERA5 reanalysis data on 37 pressure levels between $1000\,\mathrm{hPa}$ and $1\,\mathrm{hPa}$ on a regular latitude-longitude grid of $0.25^\circ \times 0.25^\circ$ are used. The data have a temporal resolution of

75 $1\,\mathrm{h}$ (Hersbach et al., 2020). Only the horizontal wind movement is considered since the convective movement of the clouds observed by specMACS is not resolved in the ERA5 data.

 The aircraft-based stereographic reconstruction of the 3-D cloud geometry is based on the observation of clouds from different perspectives by flying over them. Points on the cloud surface are identified using contrasts and then re-identified in subsequent images using the optical flow algorithm described by Lucas and Kanade (1981). Then, the points are located in 3-D

80 space using a triangular geometry as schematically shown in Fig. 1. In order to correct for cloud motion, the wind vector $\boldsymbol{v}$ at the cloud location is needed, which however is not known yet. Hence, the stereographic reconstruction is performed iteratively by first calculating the location of the point without considering the movement of the observed cloud as described in Kölling et al. (2019). Next, the horizontal wind vector is estimated at the location and time of the point found at the clouds surface by linear interpolation of the gridded reanalysis data.

85 Afterwards, the calculation of the point is repeated with the estimated cloud motion. To do so, the aircraft's locations are virtually shifted with half the wind vector each which assures that the point identified on the cloud surface is located in between the two actual locations as it is still observed under the same viewing angles. This is schematically depicted in Fig. 1. The two real aircraft positions at $P_1 = (t, \boldsymbol{x})$ and $P_2 = (t + \Delta t, \boldsymbol{x} + \Delta \boldsymbol{x})$ from which the cloud is observed are moved with the horizontal wind vector obtained from the interpolation of the ERA5 data at the initially estimated cloud top height. The initial

90 viewing points $P_1$ and $P_2$ are shifted to the points $P_1' = (t, \boldsymbol{x} + \frac{\Delta t}{2}\boldsymbol{v})$ and $P_2' = (t + \Delta t, \boldsymbol{x} + \Delta \boldsymbol{x} - \frac{\Delta t}{2}\boldsymbol{v})$. Using this modification guarantees that the reference point $P_{\mathrm{ref}}$ for the observer location remains the same. With that modified geometry, the location of the point on the cloud surface ($P_{\mathrm{CS}}$) is calculated again. Since the wind vector can only be estimated at the cloud surface

height location initially found by the method, the correction is iterated five times. Hereby, a stepwise corrected height for the estimation of an improved wind vector is used each time. It has been tested that more iteration steps do not improve the result significantly anymore.

An additional uncertainty in the stereographically estimated cloud top height is the accuracy of the ERA5 wind data which has been studied by comparisons to observations: For the surface wind field, Belmonte Rivas and Stoffelen (2019) found systematic differences of up to $0.5\,\mathrm{ms}^{-1}$ in the mean zonal and meridional components compared to the ocean vector winds of the Advanced Scatterometer (ASCAT). Hereby, the mean zonal winds are found to be overestimated whereas the mean meridional winds are found to be underestimated. Savazzi et al. (2022) and Wu et al. (2024) performed regional comparisons of the ERA5 wind. Savazzi et al. (2022) used dropsonde, radiosonde and wind lidar measurements from the EUREC$^4$A campaign to evaluate the wind bias in the lower troposphere in January and February 2020. They found root mean square errors (RMSEs) up to $2\,\mathrm{ms}^{-1}$ and a mean wind speed bias up to $-0.5\,\mathrm{ms}^{-1}$ in the lower $5\,\mathrm{km}$ of the atmosphere. Similar results with mean wind vector differences of $2.0 - 3.0\,\mathrm{ms}^{-1}$ were found by Wu et al. (2024) for comparisons of the ERA5 winds to dropsondes from the CPEX-AW campaign (Convective Processes Experiment-Aerosols and Winds) and radiosondes of the southern Great Plains (SGP) atmospheric observatory between 400 and $850\,\mathrm{hPa}$.

The maximum error in cloud top height due to wind uncertainty $\Delta h$ can be estimated using the following formula:

$$\Delta h = \frac{h_\mathrm{a} - h_\mathrm{c}}{\frac{v_\mathrm{a}}{\Delta v} - 1} \tag{1}$$

Here, $h_\mathrm{a}$ denotes the height of the aircraft, $h_\mathrm{c}$ is the cloud top height, $v_\mathrm{a}$ is the speed of the aircraft and $\Delta v$ the error in the wind speed aligned with the aircraft's flight direction. For the maximal difference of $3\,\mathrm{ms}^{-1}$ found in the mentioned studies, this would result in an additional cloud top height uncertainty of about $150\,\mathrm{m}$ if we assume an aircraft-cloud distance of $10\,\mathrm{km}$ and an aircraft speed of $200\,\mathrm{ms}^{-1}$ which are typical for the HALO aircraft. We will show later in the manuscript that such a large value is not found in comparisons using data from the EUREC4A campaign, but rather values equivalent to a wind speed bias of $0.5\,\mathrm{ms}^{-1}$.

## 3   Validation using measurements from the EUREC$^4$A campaign

The validation of the cloud motion correction on the cloud top heights derived by the stereographic reconstruction algorithm described above was conducted by considering two consecutive straight flight legs flown by HALO towards the NTAS (Northwest Tropical Atlantic Station) buoy, located at about $15\,°\mathrm{N}$ and $51\,°\mathrm{W}$ (Stevens et al., 2021), and back on 28 January 2020 during the EUREC$^4$A campaign. As described by Stevens et al. (2021), that day was associated with shallow cumulus clouds which could also be observed on the two mentioned straight legs. There were no additional cloud layers above the low shallow cumuli and hence the signal measured by specMACS and the backscatter signal measured by the WALES lidar operated simultaneously on HALO originate from the same cloud targets. Thus, the derived cloud top heights are well comparable. The straight legs were chosen for the validation because the NTAS buoy was located in the northeast of the standard "EUREC$^4$A-circle" flown by HALO (Stevens et al., 2021). Given the prevailing north-easterly trade winds in this region at that time of the

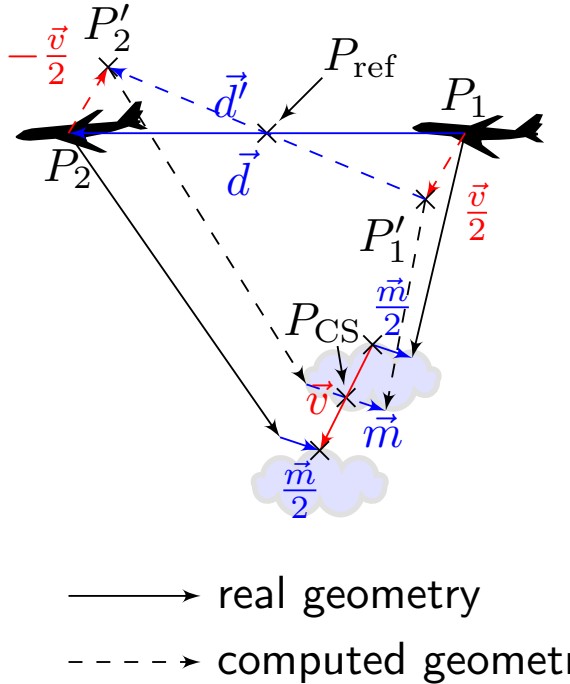

⟶ real geometry

---→ computed geometry

**Figure 1.** Tracking geometry for the location of points on the cloud surface as in Kölling et al. (2019) but with the modification for the cloud motion correction. The real geometry is shown in solid vectors, while the actually computed geometry is visualized by the dashed vectors. Hereby, $\boldsymbol{v}$ denotes the wind vector at $P_{\mathrm{CS}}$. The vector $\boldsymbol{d}$ refers to the distance vector between the two aircraft positions (typically around $200\,\mathrm{m}$) and $\boldsymbol{d'}$ is the corresponding quantity for the modified geometry. The point $P_{\mathrm{ref}}$ is the reference viewing point from which the point on the cloud $P_{\mathrm{CS}}$ is observed. Finally, $\boldsymbol{m}$ is the so-called mis-pointing vector, which takes into account that two straight lines in 3-D space do not necessarily meet. However, its length is only on the order of a few meters.

year, the first leg towards the buoy (HALO-0128_sl1) is flown against the wind while immediately afterwards the return leg (HALO-0128_sl2) is flown with the wind. The two legs were each about $20\,\mathrm{min}$ long, corresponding to a flight distance of about $270\,\mathrm{km}$. The time difference between the start of the first leg and the end of the second leg (the maximum time difference for a given position in the following analysis) was about 70 minutes. WALES measurements which are not affected by the cloud movement showed no significant changes in cloud top height during this time with median values of about $745\,\mathrm{m}$ on the first leg and $738\,\mathrm{m}$ on the second one.

Figure 2 shows the retrieved cloud top heights from the two consecutively flown straight legs projected to specMACS images of the respective flight legs. The corresponding histograms are given in Fig. 3 including the histograms and the median of the cloud top heights derived from measurements of the WALES lidar using all measurements which are referred to as most likely cloudy (Wirth, 2021). It can be seen that the stereographically derived cloud top heights on the way to the NTAS buoy and back differ by more than $600\,\mathrm{m}$ in the median without any cloud motion correction. This is due to the overestimation of the cloud

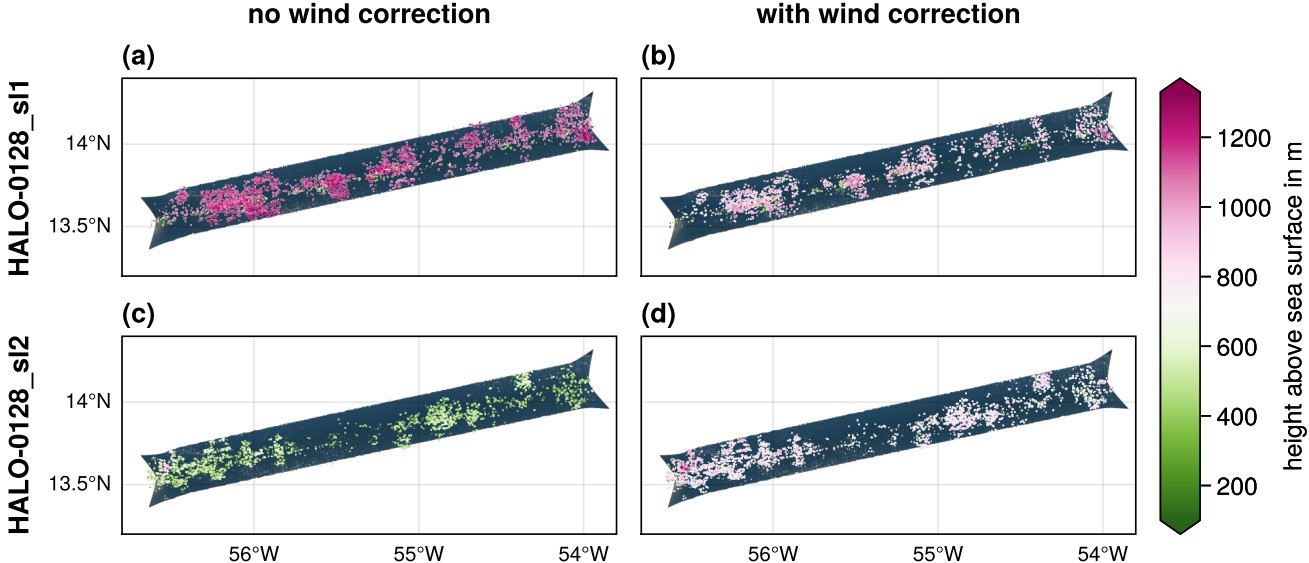

**Figure 2.** Example for the cloud motion correction of the stereographic retrieved cloud top heights of the flight segments HALO-0128_sl1 and HALO-0128_sl2 during the EUREC[4]A campaign. The legs were flown to the NTAS buoy on 28 January 2020 and directed toward the northeast such that the leg towards the buoy (sl1) was flown approximately against the wind while the wind was moving the clouds in flight direction on the way back. On the left, the stereographic retrieved cloud top heights without cloud motion correction are shown for both legs. On the right, the cloud motion correction based on the ERA5 reanalysis data is included. The background shows images from the two polarization resolving cameras of specMACS projected onto a surface at $1\,\mathrm{km}$ altitude.

top heights on the way to the buoy while flying against the wind and the underestimation on the downwind directed way back. The total horizontal wind speed was about $7\,\mathrm{ms}^{-1}$ as measured by a dropsonde launched from HALO at the end of the first leg. Including the cloud motion correction based on the ERA5 reanalysis data results in a shift of the stereographically derived cloud top heights to lower (HALO-0128_sl1) and higher (HALO-0128_sl2) values respectively such that the median values only differ by less than $60\,\mathrm{m}$. Thus, they are now in the same order of magnitude considering that not exactly the same clouds are observed on both legs and that they might develop over time. Finally, a comparison to the histograms and the median values of the lidar heights shows a much better agreement for the wind corrected heights. The difference between the stereographic derived cloud top heights and the median of the WALES measurements reduces from about $\pm300\,\mathrm{m}$ without the cloud motion correction to about $50\,\mathrm{m}$ on the way to the buoy and about $3\,\mathrm{m}$ on the way back when the cloud motion correction is included.

## 4 Accuracy estimation using realistic simulated measurements from 3-D radiative transfer simulations

To further constrain the accuracy of the presented method, realistic 3-D radiative transfer simulations with the radiative transfer model MYSTIC (Mayer, 2009; Emde et al., 2010) as part of the libRadtran radiative transfer package (Mayer and Kylling,

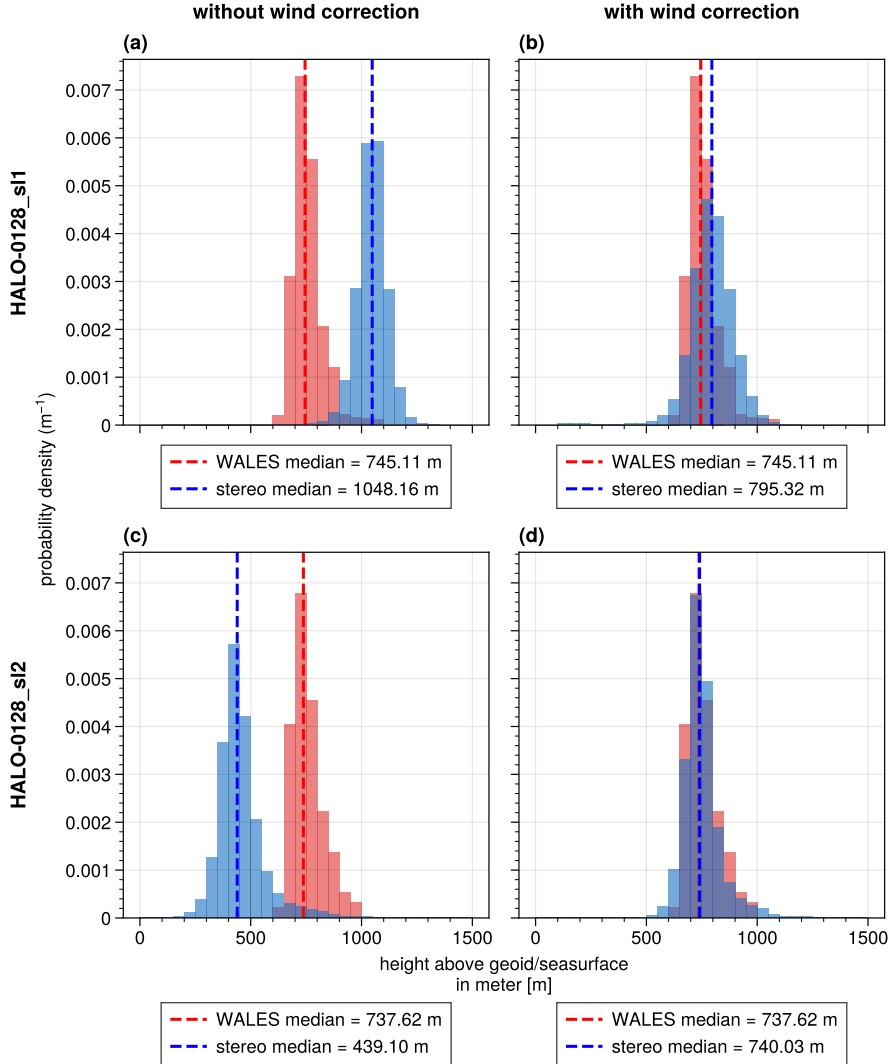

**Figure 3.** Histograms of the stereographically retrieved cloud top heights (blue) and the corresponding histograms of the WALES lidar cloud top height measurements (red) of the flight segments HALO-0128_sl1 and HALO-0128_sl2 as described in Figure 2. On the left, the histograms of the stereographically retrieved cloud top heights without cloud motion correction are shown for both legs. On the right, the cloud motion correction based on the ERA5 reanalysis data is included. Next to the median of the stereographically derived cloud top heights (blue dashed vertical line), the median of the cloud top heights derived by the WALES lidar is shown (red line).

2005; Emde et al., 2016) were performed and the stereographic reconstruction algorithm was applied to the simulated measurements of a one minute overflight over a field of large eddy simulated (LES) shallow cumulus clouds as presented by Volkmer et al. (2023b). The radiative transfer simulations were conducted for the LES cloud field with a horizontal extent of $25.6 \times 12.8\,\text{km}^2$ at a horizontal resolution of $20\,\text{m}$ and a vertical resolution of about $25\,\text{m}$. A flight altitude of $10\,\text{km}$ was

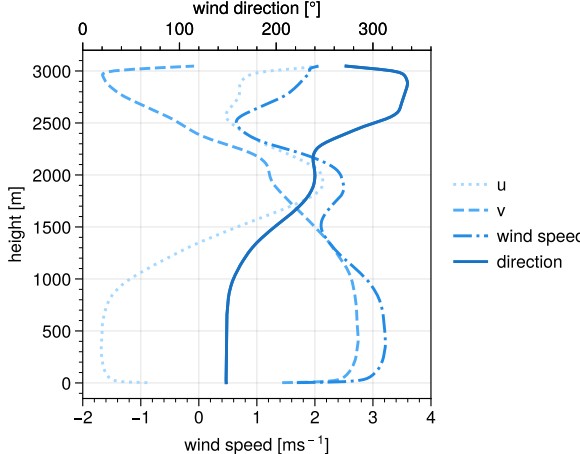

**Figure 4.** Average vertical profiles of the horizontal wind vector of the LES cloud field after a simulation time of $30\,\mathrm{s}$. The horizontal wind speed as well as the $u$- and $v$-component are shown with respect to the lower axis, while the wind direction is shown with regard to the upper axis.

assumed. Figure 4 shows the underlying model wind field as obtained from the LES simulations, indicating wind directions between approximately $140\,^{\circ}$ at low altitudes and $340\,^{\circ}$ at higher altitudes. The average wind speed is between $0.6\,\mathrm{ms}^{-1}$ at about $2500\,\mathrm{m}$ and $3.2\,\mathrm{ms}^{-1}$ between $100\,\mathrm{m}$ and $900\,\mathrm{m}$ and hence overall smaller than for the real measurements considered

before. The aircraft was simulated to fly towards the North with a horizontal aircraft alignment (all three Euler angles are zero). This corresponds to a mainly downwind flight direction for clouds lower than about $2400\,\mathrm{m}$ where the wind direction exceeds $270\,^{\circ}$.

For the comparison of the model heights to the stereographically derived cloud top heights, we used the same method as presented by Volkmer et al. (2023b). The expected cloud top heights were obtained from the first scattering events of Monte

Carlo based reference simulations, where the photons are started at the detector. For each simulated pixel, 1000 photons were computed and the scattering events of all photons not scattered at the ground were averaged to determine the height from where the optical signal originates. In Fig. 5 the expected model heights (panel a) and the corresponding stereographically derived cloud top heights including the cloud motion correction (panel c) are shown for the simulated measurements together with their respective histograms (panels b and d). On the left, the points found are projected to the measurement simulated after a flight

time of $30\,\mathrm{s}$. The flight direction is to the right of the simulated measurement. Mean cloud top heights of about $1628\,\mathrm{m}$ for the expected model heights and $1644\,\mathrm{m}$ for the stereographically derived cloud top heights are found. On the bottom of Fig. 5, the point-wise difference between the stereographic and the model heights is shown (panel e). From all point-wise differences, a mean difference of about $15\,\mathrm{m}$ between the stereo and model heights and a standard deviation of about $133\,\mathrm{m}$ is derived as shown in the corresponding histogram in Fig. 5f. Without any cloud motion correction the stereographic derived cloud top

heights for the considered LES cloud field deviated on average by $(-70 \pm 130)\,\mathrm{m}$ from the expected cloud top heights of the

model input (Volkmer et al., 2023b). Thus, the effect of the wind consideration is approximately $85\,\mathrm{m}$ for the given cloud field and aircraft direction, with a wind speed of about $2\,\mathrm{ms^{-1}}$ in flight direction. This is in accordance with the expectation from Eq. (1), resulting in a theoretical underestimation of the cloud top heights of about $84.4\,\mathrm{ms^{-1}}$, when inserting the prescribed aircraft height of $10000\,\mathrm{m}$, the mean cloud top height of about $1600\,\mathrm{m}$ and the simulated aircraft speed of $200\,\mathrm{ms^{-1}}$. It can again be observed that the point-wise differences between the stereographically derived cloud top heights and the model heights can be up to $\pm500\,\mathrm{m}$ with the positive values occurring rather where the cloud is generally lower (cloud edges and in shadow regions), while the negative ones are observed at the highest cloud tops.

## 5   Discussion

In our analysis, we could reduce the absolute mean bias in the stereographic cloud top height retrieval applied to simulated measurements from radiative transfer simulations to approximately $15\,\mathrm{m}$, when including the wind movement of the clouds. However, the standard deviation which shows the spread of the single measurements remains at about $130\,\mathrm{m}$ with and without wind correction. One possible error source, the evolution of the observed clouds, has been studied by Volkmer et al. (2023b) using a similar setup as described above but with non-developing and non-moving clouds. Hereby, an absolute mean bias of $(46\pm140)\,\mathrm{m}$ was found. While the larger mean bias might be explained by a remaining uncertainty in the wind estimation and a cancellation of the error sources, the standard deviation and hence, the error of the single measurement, remains approximately constant. One explanation for this could be the method of comparison between the stereographically determined cloud top heights and the model heights. The model heights were derived from single-scattering reference simulations and hence the simulated photons are on average scattered at an optical depth of about unity (Volkmer et al., 2023b). This is valuable because optical phenomena such as the cloudbow arise from single-scattering (Alexandrov et al., 2012) and thus, for an accurate application of the polarimetric retrieval (Pörtge et al., 2023), the cloud top heights derived by the stereographic retrieval should correspond best to the heights where the polarized cloudbow signal originates from. Moreover, in Volkmer et al. (2023b) it is argued that the stereo algorithm detects contrasts which will not be visible deeper into the cloud and that the signal smooths out when multiple scattering becomes more important. To further address this issue, we can use the model simulations to calculate the actual optical thickness at which the stereo points are found. To do so, we used the camera configuration after $30\,\mathrm{s}$ of the simulation time which is also used for the comparison between the stereo and the model heights. From Fig. 6 it can be seen that the optical thickness along the viewing directions of the given simulation time where the stereo points are found varies between about 0 and 25 with a median of 0.71 and a mean of 3.07. Further, Fig. 6c indicates that the stereo points are mostly estimated too high where the optical thickness is small (i.e. below 1) and that they are estimated too low for larger optical thicknesses. This is in accordance with the observation that the stereo points are usually estimated too low in the center of the clouds where they are found at larger optical thicknesses and too high at the cloud edges where the optical thicknesses are small (Fig. 5e and Fig. 6a).

Real measurements include further uncertainties due to limitations in the geometric camera calibration (Kölling et al., 2019, Appendix A): At a cloud-aircraft distance of $10\,\mathrm{km}$, a relative change of $0.01\,^\circ$ ($1/3\,\mathrm{px}$ for the old 2-D RGB camera and about

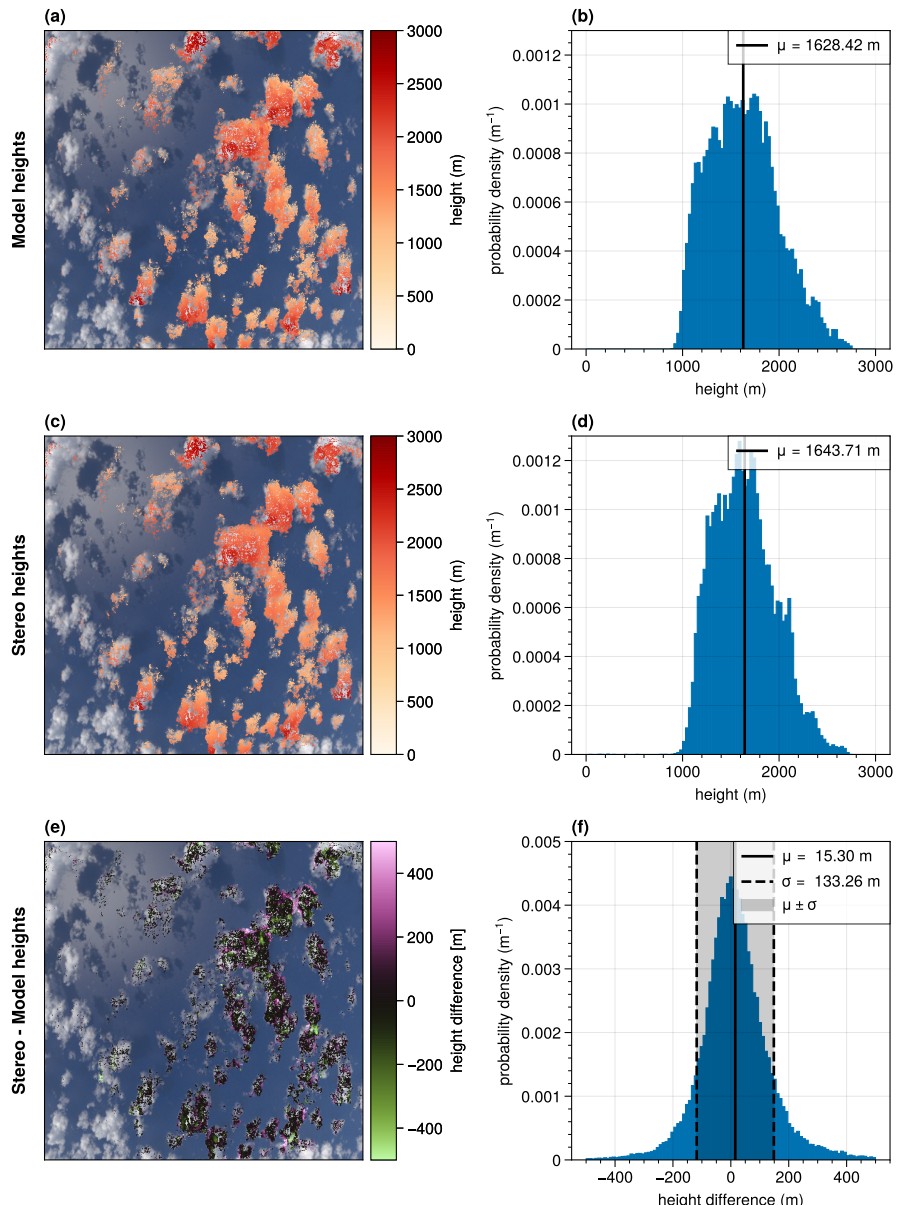

**Figure 5.** Comparison of the cloud top heights expected from the model and derived from the stereographic reconstruction algorithm. The cloud top heights as expected from the model input (Model heights) are shown in panel a (only for those points where the stereo method provided information). The stereographic derived cloud top heights can be seen in panel c. Below, the point-wise differences are shown (panel e). The derived points were projected onto the simulated RGB image and the corresponding histograms are shown in panels b, d and f. The solid vertical lines indicate the mean values. The vertical dashed lines in panel f refer to the boundaries of the $\mu \pm \sigma$ interval.

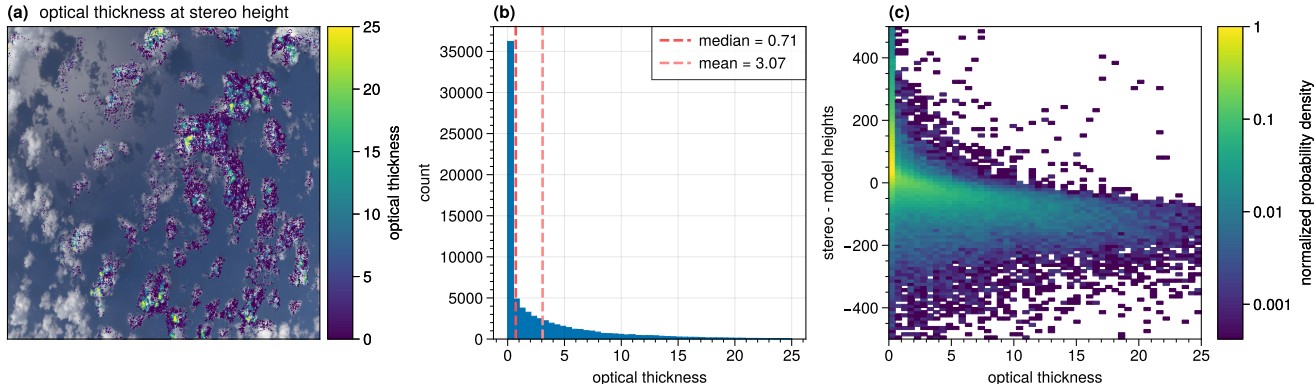

**Figure 6.** Optical thickness derived from the model input at which the stereographically derived points are found for the viewing geometry after a simulation time of $30\,\mathrm{s}$. In panel a, the optical thickness found for the stereo points is projected to the corresponding simulated RGB image. In b, the corresponding histogram of the optical thickness is shown. Panel c shows the 2-D histogram of the difference between the stereo and the model heights with regard to the optical thickness at which the stereo points are found. Note that the color scale in c is logarithmic.

$1/4\,\mathrm{px}$ for the new polarization cameras (Weber et al., 2023)) in the viewing angle results in a wrongly calculated distance
between the aircraft and the cloud of $100\,\mathrm{m}$. As described by Weber et al. (2023), the current geometric calibration of the polarization cameras of specMACS shows mean root mean square reprojection errors of the best fitting camera model of $0.18\,\mathrm{px}$ for the polLL and $0.20\,\mathrm{px}$ for the polLR cameras. Hence, this results in an additional error of about $70\,\mathrm{m}$ due to the geometric calibration which is not taken into account in this (model) study.

Stereographic cloud top height retrievals are also applied to MISR (Moroney et al., 2002; Seiz and Davies, 2006) and ASTER
(Seiz and Davies, 2006) measurements. Mitra et al. (2021) performed a detailed error analysis of the MISR cloud top heights by comparison to lidar measurements. The total error for MISR cloud top heights is estimated to $(-280 \pm 370)\,\mathrm{m}$ for optically thick, single-layered and unbroken clouds, and the precision is driven by the accuracy of the derived MISR wind speed of $3.7\,\mathrm{ms}^{-1}$ (Mitra et al., 2021). For ASTER, Seiz et al. (2006) found a systematic error of $12.5\,\mathrm{m}$ with an additional uncertainty of about $100\,\mathrm{m}$ for every $1\,\mathrm{ms}^{-1}$ uncertainty in the wind component along track of the satellite. For the future, using the wind
speed derived from the stereographic retrieval of specMACS (Kölling et al., 2019) for the correction of the cloud movement might also be possible. Hereby, the performed model simulations would allow to test the accuracy of the stereo wind and further to investigate how its usage as a cloud movement correction affects the stereo cloud top height accuracy. Depending on the performance of the derived cloud movement, this would also affect the accuracy of the stereographically derived cloud top heights. As stated above, the deviation of the ERA5 wind speed to observations is on the order of $2-3\,\mathrm{ms}^{-1}$ and a convective
development of the clouds is not taken into account. Hence, the determined stereographic cloud top height might be improved by using the stereographically derived wind vector directly.

Furthermore, Mitra et al. (2021) investigated the error of the cloud top heights with regard to the cloud height, the presence of multi-layered clouds and the optical thickness of the observed clouds. One general problem of the stereographic method is that thin clouds as well as clouds or cloud parts with a lack in contrast (e.g. homogenous cirrus clouds) are not recognized by the algorithm. Hence, in those situations only lower cloud layers will be recognized if present, and compared to e.g. lidar measurements, the cloud top heights will be estimated too low (Kölling et al., 2019). For MISR, Mitra et al. (2021) found that the lower layer is detected for an optical depth of the upper layer smaller than about 0.3. Investigations to which extent multi-layered clouds are recognized by the specMACS stereographic algorithm are also planned for the future. Moreover, clouds over ice, i.e. over sea ice in the arctic or snow covered land, are not seen by the stereographic method because of the lack in contrast between the surface and the cloud, which was frequently observed during the HALO-(AC)[3] campaign (Wendisch et al., 2024). Here, an analysis on the detection of clouds over snow or ice for the specMACS measurements would be valuable.

## 6 Conclusions

In this paper, we presented an improvement of the stereographic reconstruction method for the determination of 3-D cloud geometry from measurements of specMACS as developed by Kölling et al. (2019). The improvement of the method includes the estimation of the clouds wind movement using ERA5 reanalysis data. It could be shown that the consideration of the cloud movement with the wind is important for the estimated cloud top heights. Without any cloud motion correction the cloud top heights from two flight legs deviated by more than $600\,\mathrm{m}$ depending on the wind direction while the median of the cloud top heights derived from measurements of the WALES lidar remained approximately constant. With the consideration of the cloud motion a much better agreement between the stereographically derived cloud top heights and the ones from the WALES measurements could be achieved with median differences of about $50\,\mathrm{m}$ for the first leg and less than $3\,\mathrm{m}$ on the way back.

The improvement of the derived cloud top heights when the wind movement is considered could also be validated using realistic 3-D radiative transfer simulations following the method from Volkmer et al. (2023b). While the mean difference between the stereographically derived cloud top heights and the expected ones from the model input was $(-70 \pm 130)\,\mathrm{m}$ without the consideration of the wind movement (Volkmer et al., 2023b), the consideration of the clouds wind movement reduced the average difference to about $(15 \pm 133)\,\mathrm{m}$. Thus, while the mean bias reduces, the spread, which can be taken as a measure for the single points, is approximately constant. The method used compares the stereographically derived cloud top heights to the cloud top height at an optical thickness of approximately unity, which is the height where polarization features such as the cloudbow originate from. Hence, the accuracy of the cloud top heights from the 3-D stereographic reconstruction method which are used for the derivation of cloud microphysical properties from the polarization measurements of specMACS (Pörtge et al., 2023) can be estimated to be better than $(20 \pm 140)\,\mathrm{m}$.

While this study is based on observed and simulated shallow cumulus clouds, the performance of the stereographic retrieval including the wind correction should in future also be tested for other cloud types. In particular, the retrieval should also work in more inhomogeneous cloud fields, with cloud tops spanning larger altitude ranges. This will be investigated in the future, using

observations from past and upcoming field campaigns addressing different cloud types as well as corresponding simulated observations.

*Data availability.* The 3-D cloud geometry data from the stereographic reconstruction are available on the EUREC[4]A database on the Aeris dataserver (Volkmer et al., 2023a). The WALES (Wirth, 2021) and dropsonde data (George, 2021) are available on the database as well.

*Author contributions.* LV and TK implemented the wind correction in the stereographic reconstruction algorithm. LV applied the method to the measurements of the EUREC[4]A campaign and processed and published the data. TK, TZ and BM actively participated in the EUREC[4]A campaign and contributed all to the manuscript. TZ and BM further helped developing the implementation of the wind correction by discussing the intermediate results.

*Competing interests.* At least one of the (co-)authors is a member of the editorial board of Atmospheric Measurement Techniques.

*Acknowledgements.* The authors want to thank the EUREC[4]A campaign team for the organization, collaboration and support and especially the DLR flight operations team for planning and executing the HALO flights. Moreover, we want to thank Andreas Giez, Vladyslav Nenakhov, Martin Zöger and Christian Mallaun for providing the high time resolution BAHAMAS data necessary for the accurate location of the points on cloud surfaces in 3-D space and the evaluation of the whole flights during the campaign. Furthermore, we want to thank Veronika Pörtge and Anna Weber for their contributions in the discussions and in particular for performing the measurements and evaluation of the geometric calibration of the cameras. The data used in this publication were gathered during the EUREC[4]A field campaign and are made available via the Meteorologisches Institut, Ludwig-Maximilians-Universität München. This research has been supported by the Deutsche Forschungsgemeinschaft (DFG, German Research Foundation) within the Priority Project SPP 1294.

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
