# Peer review of "Consideration of the cloud motion for aircraft-based stereographically derived cloud geometry and cloud top heights"

_Atmospheric Measurement Techniques, 2024_

## Author Comment (AC1)

**Response to RC1 of amt-2024-19**

We want to thank the first referee of our paper for the review and comments.

The reviewer posed the following question:

*This is very nice but ignores the fact that the clouds are convective, and the vertical growth rate of the clouds can be in the order of half of the horizontal wind speed of 7 m/s in this case. An interesting question is to what extent can the remaining error be caused by the cloud's vertical growth rate?*

From a theoretical point of view, one could expect that neglecting the vertical wind speed in our consideration could explain parts of the determined uncertainty. For example, a vertical growing cloud with a growth rate of 3-4 m/s could lead to a height difference of 60 – 80 m within 20 images, which is the maximum number of frames taken into account by the stereographic reconstruction method for a single point. Hence, this could lead to "errors" in the estimation of the cloud top height of 30 – 40 m when compared to the beginning or the end of the tracking time. Here, one can ask the question what the true "cloud top height" actually is. In our model study, we chose to compare to the cloud top heights after half of the simulation time such that this effect should cancel out.

One can further argue that clouds are turbulent. In particular, the cloud edges of shallow cumulus clouds show turbulent eddies and hence, usually up- and downward movements. Thus, one probably does not observe purely growing/shrinking clouds, which will cancel out the described effect.

As described in Sec. 4 l. 154f., a large part of the uncertainty might be explained by small uncertainties in the viewing angles of the camera. This effect dominates over errors made by not considering the vertical wind.

Finally, we performed a simulation without any cloud development, and hence no wind movement (Volkmer et al., 2024, https://doi.org/10.5194/amt-17-1703-2024). We found an uncertainty of about (46 ± 140) m, which thus is in the same order as with the wind movement. We added the following paragraph from l. 155f.:

"One possible error source, the evolution of the observed clouds, has been studied by Volkmer et al. (2023b), using a similar setup as described above but with non-developing and non-moving clouds. Hereby, an absolute mean bias of (46 ± 140) m was found. While the larger mean bias might be explained by a remaining uncertainty in the wind estimation and a cancellation of the error sources, the standard deviation and hence, the error of the single measurement, remains approximately constant."

*A missing number is the simulated flight height above the simulated cloud.*

We added the missing simulated flight altitude at the beginning of Sec. 4 in line 130: "A flight altitude of 10 km was assumed."

---

## Author Comment (AC2)

**Response to RC2 of amt-2024-19**

We want to thank the second referee for reviewing our manuscript. The reviewer posed some minor comments which we have addressed as described below:

**Wind correction based on ERA5 reanalysis data**

Please, add also *Pcs* in the text.

p. 3, line 90: "of the point on the cloud surface (Pcs)"

to Figure 1: In the printed version the clouds are quite faint.

We applied the suggested change in l. 90/91 and enlarged the opacity of the clouds in Fig. 1.

**Validation using measurements from EUREC4A campaign**

Figure 2 should be enlarged, it hard to see.

The figure size has been enlarged as suggested.

p.4, line 109: "no significant change" To be more precise, a value should be given here.

We added the median values of the flight legs from the lidar in the suggested sentence: "WALES measurements which are not affected by the cloud movement showed no significant changes in cloud top height during this time with median values of about 745 m on the first leg and 738 m on the second one."

Figure 3: Could you include the PDF of the WALES measurement, the comparison would be benefit from it.

The WALES PDFs were added to Fig. 3.

**Accuracy estimation using realistic simulated measurements from 3-D radiative transfer simulations**

p.7, line 133. Can you add an explanation why the used wind speed in the simulations is chosen lower than the real measurement before?

The wind speed of the simulations was taken as obtained from the LES simulations. The simulations were initialized using data from the EUREC$^4$A campaign but afterwards not forced any further. Since we used the second-by-second output from the LES for our radiative transfer simulations, the wind is prescribed by the output, and thus, was not chosen. We changed l. 130f. as following to emphasize that the wind field was obtained from the simulations and was not chosen: "Figure 4 shows the underlying model wind field as obtained

from the LES simulations, indicating wind directions between approximately 140° at low altitudes and 340° at higher altitudes."

**Conclusion**

To the conclusion: It should be mentioned that further studies with more inhomogeneous cloud state should be added. Since the correction will be even more necessary.

To address this comment, we added the following paragraph at the end of the paper:

"While this study is based on observed and simulated shallow cumulus clouds, the performance of the stereographic retrieval including the wind correction should in future also be tested for other cloud types. In particular, the retrieval should also work in more inhomogeneous cloud fields, with cloud tops spanning larger altitude ranges. This will be investigated in the future, using observations of from past and upcoming field campaigns addressing different cloud types as well as corresponding simulated observations."

---

## Author Response (AR2)

**Response to referee #3 of amt-2024-19**

We want to thank the third referee of our paper for the review and comments.

The reviewer addressed the following minor comments:

Line 6ff: Please write out all abbreviations when introduced. The Paper might as well be read by people from the photogrammetry community who are not familiar with abberivations such as ECMWF and ERA5.

Line 66: Please mention that ERA5 is produced by ECMWF. In the abstract you only mention ECMWF as a source of model wind data.

We wrote out ECMWF in the abstract (l.6) and added a description of ERA5 in the introduction as well as that it is produced by ECMWF (l.66f): "the fifth generation European Centre for Medium-Range Weather Forecasts (ECMWF) atmospheric reanalysis (ERA5, Hersbach et al., 2020)"

Line 71ff: The spatial and temporal resolution of the ERA5 reanalysis data may be too low to resolve local wind phenomena, e.g. in coastal areas. Also the general accuracy of ERA5 wind data might be a factor that could affect your results of the method. Please add some discussion.

We added some discussion regarding the accuracy of the ERA5 wind data at the end of Sec. 2 as well as in a new separate discussion section (Sec. 5).

Line 97: Please add the location of the NTAS buoy in geographic coordinates and the length of the flight legs.

We added the location of the NTAS buoy in l. 116: "The validation of the cloud motion correction on the cloud top heights derived by the stereographic reconstruction algorithm described above was conducted by considering two consecutive straight flight legs flown by HALO towards the NTAS (Northwest Tropical Atlantic Station) buoy, located at about 15°N and 51°W (Stevens et al., 2021), and back on 28 January 2020 during the EUREC⁴A campaign."

Further we added the length of the flight segments in line 124f.: "The two legs were each about 20 min long, corresponding to a flight distance of about 270 km."

Line 100: Was the earth's surface also visible and recognized by the stereographic reconstruction algorithm?

The earth's surface might be tracked by the algorithm if it has a high enough contrast, which can for example be the case for cloud shadows visible on the surface. However, it is actually removed by filtering out points below a given height threshold. For the EUREC⁴A campaign, we used a threshold of 100 m above sea surface since the flights were conducted over the ocean only. For other campaigns, we are also able to use digital elevation models as a reference. We tried to make this a little bit clearer by rephrasing the text from line 117f, since the sentence regarding the contrast might be a little bit misleading as it should be clear that contrasts are

needed for the stereographic reconstruction: "As described by Stevens et al. (2021), that day was associated with shallow cumulus clouds which could also be observed on the two mentioned straight legs. There were no additional cloud layers above the low shallow cumuli and hence the signal measured by specMACS and the backscatter signal measured by the WALES lidar operated simultaneously on HALO originate from the same cloud targets."

Line 179: The differences (50 m and 3 m) are different from those specified in line 124 (40 m and 5 m). Shouldn't they be the same?

Yes, thanks for recognizing this. We corrected the numbers.

Additional remarks:

1: To further improve the method and derive the 'actual' wind, wouldn't it be a good idea to - from time to time - fly a short segment (e.g. 1 minute) into one direction, return on the same path and then continue the flight? By doing so, a wind speed can be determined at which the stereographically derived cloud top hights are the same for the forward and return segment. You could test this approach by evaluating the two flight legs at a location near the NTAS buoy.

Yes, this could be an idea, at least we could try to evaluate the turning points as suggested for the location near the NTAS buoy for the data available from previous campaigns. However, it might be a little hard to achieve the actual idea with the 1 min flight legs, since it takes about 7 min for HALO to do a 360° procedure turn, hence it would be a very time consuming and expensive study. But still, there are probably a lot of data from past campaigns which include aircraft turns as presented here with the NTAS buoy example.

2: Please add a little more discussion about the limitations of your method (multi-layer clouds, accuracy of model winds, ...)

We added some discussion on the accuracy of the ERA5 wind in Sec. 2 and its implication to the height accuracy of the method.

Furthermore, we added a Discussion Section which includes parts of the previous Sec. 4 and some further discussion about the origin of the uncertainty that we found here.

**Response to referee #4 of amt-2024-19**

We want to thank Arka Mitra for reviewing our paper and the comments he made.

The reviewer posed the following issues:

1) (Lines 155-170): About the issue with the 10 km gap between the flights and cloud being attributed for the unwavering random error, a case can be made for a sensitivity analysis that confirms this hypothesis. Varying the height could allow the authors to constrain the uncertainty and judge whether any other underlying sources of error are being overlooked. However, that may be too much work, I don't know.

What makes me think this is because having been involved in a detailed investigation of MISR CTH errors (see Mitra et al 2021), I can inform the authors that the numbers they quote on MISR's geo-registration errors were only true for older MISR data records. With sub-pixel adjustments now, MISR image geo-registration error is estimated to be 0.05 ± 0.25 pixels, translating to height errors of ~30 ± 140 m (see Davies et al. 2007 or Jovanovic et al. 2007). Now that is eerily like what the authors themselves have found. Maybe address this?

I am somewhat hesitant to believe that this flight-based technique and a satellite sensor has similar geo-registration uncertainties, which makes me suspect that other sources of errors might be at play. is very nice but ignores the fact that the clouds are convective, and the vertical growth rate of the clouds can be in the order of half of the horizontal wind speed of 7 m/s in this case. An interesting question is to what extent can the remaining error be caused by the cloud's vertical growth rate?

Thanks for pointing this out. We noticed now, that the error we get from the simulations is actually not due to the geometric calibration of the camera because the simulated viewing directions are the ones obtained from the camera model which is used by the stereo algorithm.

Hence, we added some discussion concerning the origin of the error found here, which is most likely due to the method used for the comparison between the stereo and the model heights. We used reference simulations of single scattered photons to determine the expected cloud top height. One reason for that is that one main usage of the stereo heights is the polarimetric retrieval for cloud microphysics using polarized observations of the cloudbow (see Pörtge et al., 2023). The cloudbow is an optical phenomenon originating from an optical thickness of about 1. Furthermore, we argued before that the algorithm detects contrasts which are not visible deeper into the cloud and that the signal smooths out when multiple scattering becomes more important (Volkmer et al., 2023). To address this issue, we added some analysis showing from which optical depth the stereo height signal actually comes from finding optical depths between 0 and 25 and discussing the implications for our findings: the uncertainty is valid for the expected cloud top heights at an optical depth of about unity.

2) Also, I am intrigued by the change in "sign" of the bias before and after wind-rectification. A negative error (such as -70 ± 130 m) is intuitive to a stereo height. In fact, for optically thicker clouds it tends to be negative numbers closer to zero, but negative, nonetheless. However, I would be interested to listen to the authors' explanations as to why they think after wind correction, that number is positive (i.e., stereo detects a height higher than "truth" or in other words, no correlation to cloud opacity anymore). Again, I leave it to the authors' discretion if they want to address this point. missing number is the simulated flight height above the simulated cloud.

As implied by the above mentioned analysis concerning the optical depth from which the stereo signal comes from, we find that positive differences mainly occur where the stereo point is found at an optical depth of less than 1. Negative anomalies with regard to the model heights are found more in the center of the clouds at large optical thicknesses. On average, the fact that many points are found at low optical thicknesses leading to positive differences, leads probably to the overall positive bias.